# Structural, Spectroscopic, and Thermal Decomposition Features of [Carbonatotetraamminecobalt(III)] Iodide—Insight into the Simultaneous Solid-Phase Quasi-Intramolecular Redox Reactions

Kende Attila Béres [1,2,*], Fanni Szilágyi [3], Zoltán Homonnay [2], Zsolt Dürvanger [4,5], Laura Bereczki [1,6], László Trif [1], Vladimir M. Petruševski [7], Attila Farkas [8], Niloofar Bayat [1,9] and László Kótai [1,10,*]

1 Research Centre for Natural Sciences, Institute of Materials and Environmental Chemistry, Magyar Tudósok krt. 2, 1117 Budapest, Hungary
2 György Hevesy PhD School of Chemistry, Institute of Chemistry, ELTE Eötvös Loránd University, Pázmány Péter s. 1/A, 1117 Budapest, Hungary
3 Production Division (BAY-PROD), Bay Zoltan Ltd. for Applied Research, 1 Kondorfa, 1116 Budapest, Hungary
4 ELKH-ELTE Protein Modelling Research Group, Pázmány Péter s. 1/A, 1117 Budapest, Hungary
5 Structural Chemistry and Biology Laboratory, Institute of Chemistry, ELTE Eötvös Loránd University, Pázmány Péter s. 1/A, 1117 Budapest, Hungary
6 Research Centre for Natural Sciences, Centre for Structural Science, Magyar Tudósok krt. 2, 1117 Budapest, Hungary
7 Faculty of Natural Sciences and Mathematics, Ss. Cyril and Methodius University, 1000 Skopje, North Macedonia
8 Department of Organic Chemistry and Technology, Faculty of Chemical Technology and Biotechnology, Budapest University of Technology and Economics, Műegyetem rkp. 3, 1111 Budapest, Hungary
9 Department of Analytical and Inorganic Chemistry, Budapest University of Technology and Economics, Műegyetem rakpart 3, 1111 Budapest, Hungary
10 Deuton-X Ltd., Selmeci u. 89, 2030 Érd, Hungary
* Correspondence: beres.kende.attila@ttk.hu (K.A.B.); kotai.laszlo@ttk.hu (L.K.)

**Abstract:** [$\kappa^2$-O,O'-Carbonatotetraamminecobalt(III)] iodide, or [Co(NH$_3$)$_4$CO$_3$]I, named in this paper as compound **1,** was prepared and characterized comprehensively with spectroscopic (IR, Raman and UV) and single-crystal X-ray diffraction methods. Compound **1** was orthorhombic, and isomorphous with the analogous bromide. The four ammonia ligands and the carbonate anion were coordinated to the central cobalt cation in a distorted octahedral geometry. The carbonate ion formed a four-membered symmetric planar chelate ring. The complex cations were bound to each other by N-H···O hydrogen bonds and formed zigzag sheets via an extended 2D hydrogen bond network. The complex cations and iodide ions were arranged into ion pairs and each cation bound its iodide pair through three hydrogen bonds. The thermal decomposition started with the oxidation of the iodide ion by Co$^{III}$ in the solid phase resulting in [Co(NH$_3$)$_4$CO$_3$] and I$_2$. This intermediate Co$^{II}$-complex in situ decomposed into Co$_3$O$_4$ and C-N bond containing intermediates. In inert atmosphere, CO or C-N bond containing compounds, and also, due to the in situ decomposition of CoCO$_3$ intermediate, Co$_3$O$_4$ was formed. The quasi-intramolecular solid-phase redox reaction of [Co(NH$_3$)$_4$CO$_3$] might have resulted in the formation of C-N bond containing compounds with substoichiometric release of ammonia and CO$_2$ from compound **1**. The C-N bond containing intermediates reduced Co$_3$O$_4$ into CoO and Co, whereas in oxygen-containing atmosphere, the end-product was Co$_3$O$_4$, even at 200 °C, and the endothermic ligand loss reaction coincided with the consecutive exothermic oxidation processes.

**Keywords:** ammine; carbonate; coordination; spectroscopy; thermal analysis; crystal structure; correlation analysis; cobalt; hydrogen bond

## 1. Introduction

Solid-phase quasi-intramolecular redox reactions of the transition metal complexes having redox active ligands, cations and anions ensure an excellent route for the preparation of nanometer structured transition metals, and their alloys, oxides or nitrides [1–20]. In order to prepare effective catalysts for Fischer–Tropsch syntheses and photodegradation reactions, a number of cobalt oxide-based catalysts have also been prepared in this way from cobalt(III) ammine complexes, including [carbonatotetraamminecobalt(III)] salts [1,12,21–24]. To prepare various soluble carbonatotetraamminecobalt(III)-salts, the direct metathesis reactions of [carbonatotetraamminecobalt(III)]iodide and Ag- or Pb$^{II}$-salts (due to low solubility of AgI and PbI$_2$, $L = 8.5 \times 10^{17}$ and $1.4 \times 10^{-8}$, respectively) are promising alternatives. Surprisingly, [Co(NH$_3$)$_4$CO$_3$]I reacts with lead thiocyanate resulting in [carbonatotetraamminecobalt(III)] hydrocarbonate instead of the expected thiocyanate salt [22]. It is quite surprising, because of the existence of stable thiocyanate complexes of cobalt [25], however, this reaction route ensures a convenient way to prepare new [carbonatotetraamminecobalt(III)] salts with the interaction of the formed hydrogen carbonate compound with acids stronger than carbonic acid. These [carbonatotetraamminecobalt(III)] complexes are possible precursors of various simple or mixed cobalt oxide catalysts. Among them, only the permanganate [21,22] and sulfate compounds [24] have been exploited. In order to find new precursors, we screened these compounds to find one with an easily eliminable anion to prepare co-oxides. Only four [carbonatotetraamminecobalt(III)] complexes have known structures ([Co(NH$_3$)$_4$CO$_3$]X, X = Br [26,27], NO$_3$ [28–30], $\frac{1}{2}$ SO$_4$ [31], and ClO$_4$ [19]), and among the four halogenide derivatives (X = F [31], Cl, Br or I [32]), the thermal decomposition pathways were only clarified for the chloride and bromide salts [33]. In the case of [$\kappa^2$-O,O′-carbonatotetraamminecobalt(III)] iodide, ([Co(NH$_3$)$_4$CO$_3$]I, compound **1**), iodine vapor evolution and non-stoichiometric thermal decomposition reaction could be observed. Generally, in the thermal decomposition process of Co$^{III}$-ammine complexes, the dissociation of a transient state [Co$^{III}$-NH$_2$$^{d-}$···H$^{d+}$] into H$^+$ and NH$_2$$^-$ accelerates a concomitant electron transfer from the NH$_2$$^-$ moiety to the Co$^{III}$ with a subsequent N$_2$ formation [34]. However, due to the presence of iodide ion [35–37], the reducing agent may not only be the coordinated ammonia but the iodide as well. Therefore, the study of the iodide compound is especially interesting. Although the IR spectrum of compound **1** is already known [38,39], the UV, the far-IR and the Raman measurements have not yet been reported. Therefore, in order to clarify the nature of the redox reactions in compound **1** during its thermal decomposition, including the role of ammonia ligands, to obtain nanosized Co-oxides, we studied the thermal decomposition of compound **1** using TG-MS, DSC and powder XRD methods. In order to evaluate the role of structural features in the thermal decomposition process, single crystal XRD structure determination and detailed spectroscopic (IR, far-IR, Raman and UV) analyses were also conducted.

## 2. Results and Discussion

### 2.1. Preparation and Properties of Compound **1**

[Carbonatotetraamminecobalt(III)] iodide was prepared according to a method slightly different from that given by Jörgensen [32]. Since the solubility of compound **1** in water (1.6 g/100 mL) is lower than that of the corresponding nitrate (6.7 g/100 mL), it can easily be separated in the reaction of [carbonatotetraamminecobalt(III)] nitrate with ammonium iodide [32] with excellent yield. The starting [carbonatotetraamminecobalt(III)] nitrate salt was prepared from cobalt(II) nitrate hexahydrate dissolved in aq. ammonia with subsequent addition of ammonium carbonate and dilute (3%) aq. hydrogen peroxide. The aq. solution of [carbonatotetraamminecobalt(III)] nitrate was reacted with the aq. solution of ammonium iodide at ~40 °C. The solution was cooled and left to crystallize overnight, then compound **1** was isolated at 93.3% yield.

The same product was obtained with the use of another method given by Jörgensen [32], where the starting material was cobalt(II) iodide prepared in situ from cobalt(II) carbonate and hydroiodic acid. Cobalt(II) iodide was complexed with aq. ammonia and oxidized by

air for 2.5 h in the presence of excess ammonium carbonate. The insufficient ammonium carbonate addition resulted in the separation of [hexaamminecobalt(II)] iodide.

The products of the reactions mentioned above were a carmine-red mass of needle-like crystallites. Compound **1** was stable in air at room temperature. The compound was moderately soluble in water; its solubility value was 1.6 g/100 mL at room temperature. Compound **1** was insoluble in organic solvents such as methanol, ethanol, chloroform and carbon tetrachloride. The saturated aqueous solution of compound **1** was red with a pH value of 5.3. Compound **1** was orthorhombic, isomorphous with the bromide analogue [26]. DSC results showed the lack of polymorph phase transition of compound **1** between $-150\,°C$ and its thermal decomposition point (Figure S1).

Compound **1** contained non-coordinated iodide, thus, the iodide content of compound **1** was determined gravimetrically after precipitation with silver nitrate. The determination of cobalt, ammonia and carbonate contents of compound **1** was possible only after the decomposition of the stable complex cation in compound **1**. During a usual alkaline hydrolysis [24], the cobalt(III) content would oxidize iodide ions into iodine, and the elementary iodine would react with ammonia, with harmful $NI_3$ and HI formed. In order to avoid iodine formation, the iodide content of compound **1** had to be eliminated before the hydrolysis of compound **1**. Therefore, a slight excess of $AgNO_3$ solution was used to eliminate iodide ions as insoluble AgI, then the obtained solution was boiled with NaOH. The ammonia content was measured gravimetrically with $H_2PtCl_6$ as $(NH_4)_2PtCl_6$ or by absorption in 0.1 M HCl with back-titration of the excess HCl [12,13]. On acidifying the [carbonatotetraamminecobalt(III)] nitrate solution with sulfuric acid, and boiling that, the evolved $CO_2$ was measured gravimetrically as $BaCO_3$ [24]. To determine the cobalt(III) content, the residual mother liquor was treated with sodium peroxydisulfate and NaOH, and the mass of the obtained brown precipitate was measured as $Co_3O_4$ after heating at $950\,°C$ for 2 h [24].

Based on the analysis, compound **1** proved to be anhydrous, as also found by Jörgensen [32].

### 2.2. Crystal Structure Features of Compound **1**

A single-crystal X-ray measurement was performed on the purple needles of compound **1** at 303 K. The tetraammine(carbonato-$\kappa^2$-O,O')cobalt(III) iodide crystallized in the orthorhombic crystal system, in the *Pnma* space group. The cell parameters are given in Table S1: $a = 17.7359(2)$ Å, $b = 7.77940(10)$ Å, $c = 6.82520(10)$ Å, $Z = 4$, and $d_{calc.} = 2.215$ g cm$^{-3}$. The space groups *Pmnb* and *Pcmn* (no. 62 each) were given by Barclay [27] and Haagensen [26] for $[Co(NH_3)_4CO_3]Br$ (compound **2**), thus, compounds **1** and **2** were isomorphous. Crystal data and details of the structure refinement are listed in Table S1**.** The calculated powder X-ray diffractogram was identical to the experimentally found one (Figure S2).

The four ammonia ligands and the carbonate anion were coordinated to the central cobalt cation in a distorted octahedral geometry (Figure 1). The bond lengths and angles are given in Tables S2 and S3, respectively. The coordination octahedron was quite distorted at the chelate part since the O2-Co1-O2#1 bond angle was only 68.3° while the O2-Co1-N1 bond angle was 98.6°. The equatorial Co-N distances were equal, 1.959(3) Å, where the two trans ammonia ligand distances from the central $Co^{III}$ ion were 1.957(5) and 1.953(4) Å. The trans-effect of carbonate ion on the opposite $NH_3$ ligands in the structure of compound **2** mentioned by Barclay could not be found in the structure of compound **1**. This agreed with the conclusion of Shi [40], who found that the carbonate trans-effect in the structure of compound **2** was negligible. The Co-N distances in compound **1** were comparable with those found in $[Co(NH_3)_5CO_3]I$ (1.90-1.98 Å) [41] and $[Co(NH_3)_6]I_3$ (average 1.96 Å) [42]. The carbonate ion formed a four-membered symmetric planar chelate ring. The symmetric carbonate ion geometry deviated somewhat from the ideal one and one short (double-bond character) and two long (single bond character) C-O bonds were present, which could be distinguished in the vibrational spectra (see below). The C=O and C-O···(Co) carbon oxygen distances were 1.227 Å and 1.308 Å, comparable with the values found for the

bromide analogue (compound **2**), 1.237 and 1.336 Å, respectively. The O-C=O and O-C-O bond angles of iodide (124.5(2)° and 111.0(4)°, respectively) fell between the values found for bromide by Barclay [27] and Shi [40] (124.8/123.1° and 110.4/112.7°, respectively).

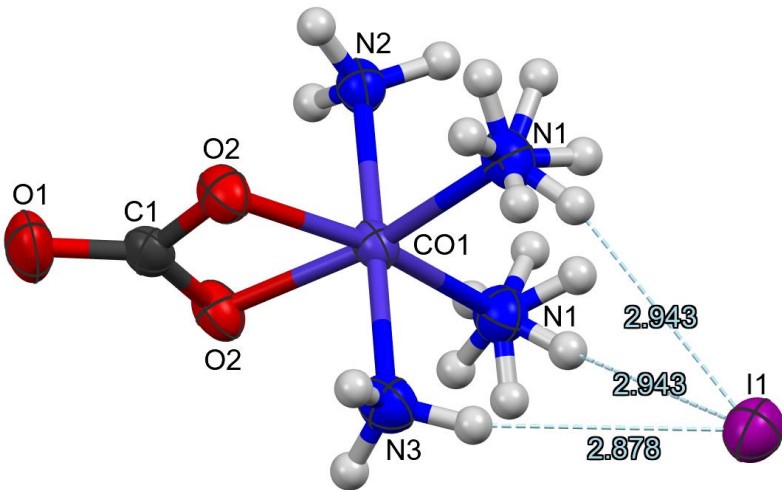

**Figure 1.** Structure of compound **1** at 100 K (thermal ellipsoids are drawn at the 50% probability level, hydrogen bonds between the ammonia ligands and the iodide anion are drawn by blue dashed lines).

This shows that the chemical nature of the halogenide ions in compounds **1** and **2** have no important role in the geometry of the carbonate ion, thus, they had ionic character. The perchlorate analogue of compound **1**, $[Co(NH_3)_4CO_3]ClO_4$, was an anhydrous compound (orthorhombic, *Pnma*) with analogous symmetry relations regarding the co-ordinated ammonia ligands. The equatorial Co-N distances were the same, 1.9590(19) Å, and the two axial Co-N distances were close to those in compound **1**, 1.947(3) Å and 1.952(4) Å for Co-$N_{ax1}$ and Co-$N_{ax2}$, respectively. Similarly, the two Co-O distances for the carbonate oxygens were the same (1.303(2) Å), and the non-coordinated C=O bond length (1.250(4) Å) was shorter than the coordinated C-O bonds [19]. The presence of 0.5 and 1.5 crystallization water in the nitrate and sulfate analogues, $[Co(NH_3)_4CO_3]NO_3.0.5H_2O$ and $[Co(NH_3)_4CO_3]_2SO_4.3H_2O$, respectively, however, generated the possibility of new hydrogen bonds. Racemic conglomerate crystal formation of the nitrate salt with clavic dissymmetry were found by Bernal [29]. All four Co-N distances were different in the two dissymmetric cationic units (for Co-$N_{ax1}$:1.963(1)/1.969(1) Å, for Co-$N_{ax2}$:1.952(1)/1.955(1) Å, for Co-$N_{eq1}$:1.948(1)/1.942(1) Å, and for Co-$N_{eq2}$:1.951(1)/1.949(1) Å, respectively). The two Co-O distances in each cationic component were found to be different, 1.903(1)/1.913(1) Å and 1.896(1)/1.916(1) Å, respectively. One of the carbonate ion symmetries, (where the bond length for (Co⋯)O-C was 1.305(2) Å, and for C=O was 1.227(2) Å) was similar to that in compound **1**, while the carbonate ion in the second kind of cation was distorted completely (the C-$O_{coord1}$, C-$O_{coord2}$ and C=O distances were 1.297(2) Å, 1.322(2) Å and 1.222(2) Å, respectively). The higher number of hydrogen bonds caused complete loss of symmetry in the cations of the sulfate compound. All three pairs of carbon–oxygen, all four pairs of Co-N and the two pairs of Co-$O_{coord}$ bond lengths were different (C-$O_{coord1}$, C-$O_{coord2}$, C=O, Co-$N_{eq1}$, Co-$N_{eq2}$, Co-$N_{ax1}$, Co$N_{ax2}$, Co-$O_{coord1}$, and Co-$O_{coord2}$ distances were 1.315(5)/1.317(4), 1.319(5)/1.319(4), 1.226(4)/1.234(4), 1.963(3)/1.960(3) Å, 1.957(3)/1.967(4) Å, 1.955(3)/1.942(3) Å, 1.944(3)/1.957(4) Å, 1.897(2)/1.905(2) Å, and 1.915(2)/1.1919(2) Å, respectively).

The complex cations and the iodide anions in the lattice were arranged in rows running parallel to the *c* crystallographic axis (Figure 2). The tetraammine(carbonato-$\kappa^2$-O,O′)cobalt(III) complex cations were bound to each other by N-H⋯O hydrogen bonds and formed zigzag sheets (Table S4).

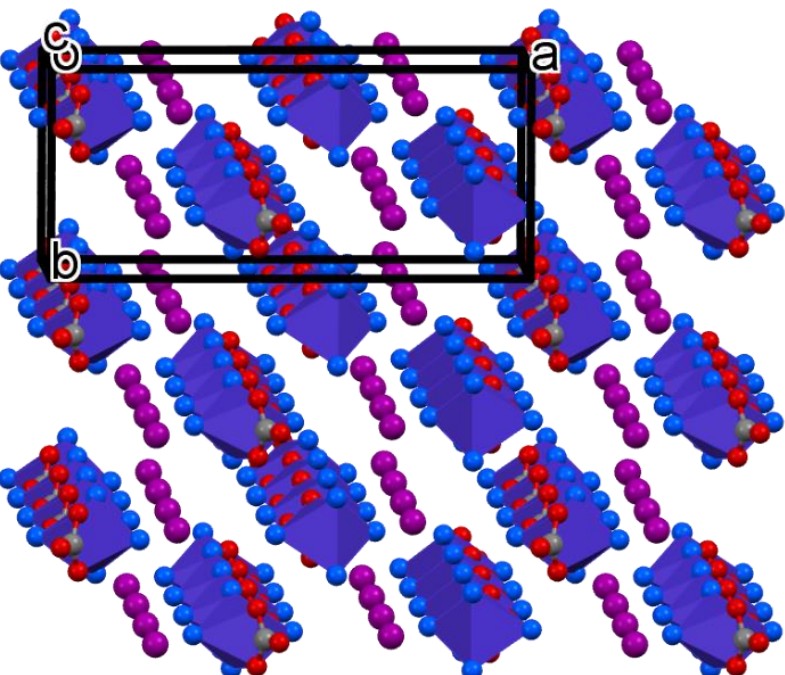

**Figure 2.** Packing arrangement in the crystal lattice of [Co(NH$_3$)$_4$CO$_3$]I (a, b, c are axes marks and o is the origin).

O1 formed five hydrogen bonds with the ammonia ligands of three other complexes and O$_2$ formed two hydrogen bonds with one neighbouring complex. The complex cations and iodide ions were arranged to ion pairs and each cation bound its iodide pair through three hydrogen bonds. The Kitaigorosdkii packing index was 68.2%.

### 2.3. IR, Raman and UV Spectroscopic Characterization of Compound 1

The IR bands belonging to the internal vibrational modes of the complex carbonatotetraamminecobalt(III) iodide have already been published and assigned earlier [38,39,43], however, neither Raman spectroscopic results nor unit-cell group vibrational analysis data were available for compound **1**. Therefore, room temperature and low-temperature (−150 °C) Raman, as well as IR and far–IR measurements were performed by us. The total number of the vibrational modes in [Co(NH$_3$)$_4$CO$_3$] complex cation of compound **1** was also predicted by the correlation method.

### 2.3.1. Correlation Analysis of Compound 1

The unit cell of compound **1** consisted of carbonatotetraamminecobalt(III) cations (i.e., three kinds of non-equivalent ammonia molecules (1 + 1 + 2), one carbonate anion and one type of Co$^{III}$ ion) as well as one type of iodide ion. The vibrations of the above building blocks were treated separately.

### 2.3.2. The Ammonia Ligands in Compound 1

There were, as briefly mentioned above, three crystallographically non-equivalent ammonia ligands in compound **1**. Among these, two ammonia molecules from the formula unit were in positions of trivial symmetry (type *I*). The other two ammonia molecules were crystallographically different and both were located on xz planes of symmetry (type *II*). Thus, the number of modes (external or internal) for the two crystallographically different ammonia molecules (type *II*) needed to be doubled. The correlation tables for the internal and external modes of ammonia molecules are given in Figure 3.

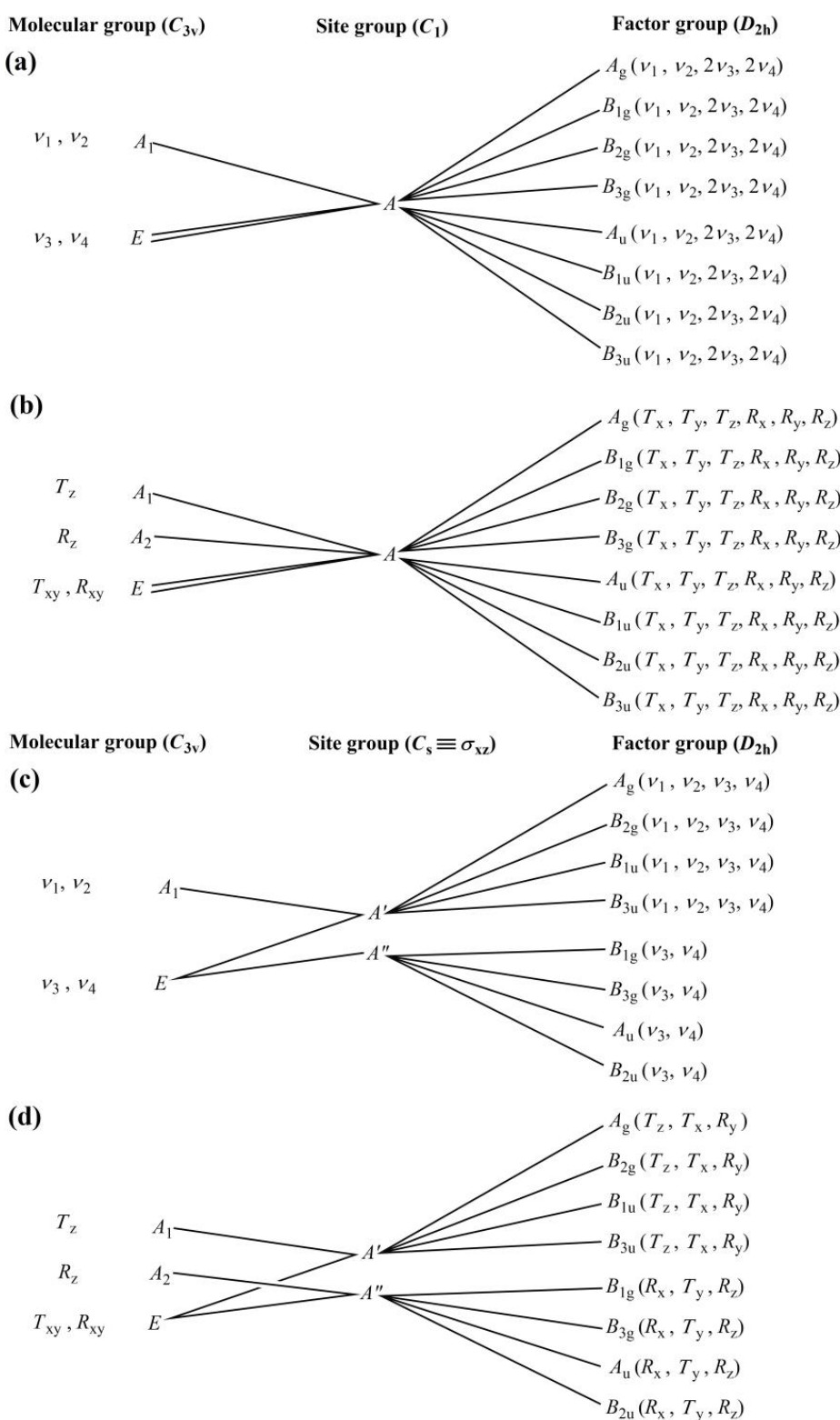

**Figure 3.** Factor group analysis of (**a,c**) internal, and (**b,d**) external, modes of ammonia molecules in $[Co(NH_3)_4CO_3]I$. One kind of the three crystallographic types of ammonia molecules is on positions of trivial symmetry (type *I*, $C_1$)l (**a,b**), whereas the other two crystallographic kinds of ammonia molecules are on *xz* planes of symmetry (type *II*, (**a,c**)).

For the ammonia molecules of type *I*, $4 \times 6 = 24$ IR and $4 \times 6 = 24$ Raman active internal modes were expected in every species ($A_u$, $B_{1u}$, $B_{2u}$, $B_{3u}$ and $A_g$, $B_{1g}$, $B_{2g}$, $B_{3g}$, respectively,

2 singlets ($\nu_1$, $\nu_2$) and 2 doublets ($\nu_3$, $\nu_4$) for each). The number of external modes were $4 \times 3$ rotational and $4 \times 3$ translational modes in the same species ($A_u$, $B_{1u}$, $B_{2u}$, $B_{3u}$, and $A_g$, $B_{1g}$, $B_{2g}$, $B_{3g}$, respectively). This meant that 24 IR and 24 Raman bands belonging to external modes were to be expected at most. Band overlapping was very likely.

The type *II* ammonia molecules showed a different picture (Figure 3). There were 2 bands both in the IR and Raman of $\nu_1$ and $\nu_2$ ($B_{1u}$ or $B_{3u}$, and $A_g$ and $B_{2g}$ species, respectively), whereas the $\nu_3$ and $\nu_3$ modes appeared in all four species belonging to IR ($A_u$, $B_{1u}$, $B_{2u}$, $B_{3u}$) and Raman ($A_g$, $B_{1g}$, $B_{2g}$, $B_{3g}$) active modes. This meant that $2 \times 2 \times 4 + 2 \times 2 \times 2$ IR and the same Raman internal modes ($2 \times 2\nu_1$, $2 \times 2\nu_2$, $2 \times 4\nu_3$ and $2 \times 4\nu_4$) were expected (Figure 3). The $B_{1u}$, $B_{3u}$, $A_g$ and $B_{2g}$ species contained 2 translational ($T_x$, $T_z$) and 1 rotational ($R_y$) mode each, both in the IR and Raman, whereas the $A_u$, $B_{2u}$, $B_{1g}$ and, $B_{3g}$ species each contained the other two ($R_x$, $R_z$) rotational and $T_y$ translational modes, both in the IR and Raman. For two different types of such ammonia molecules, the number of modes (external or internal) needed to be doubled. According to these, $2 \times 4 \times 3$ translational and $2 \times 4 \times 3$ rotational modes were expected both in the IR and Raman spectra.

The symmetry of the carbonate ion changed from $D_{3h}$ to $C_s$. The $\nu_2$ internal and $T_y$ translational and $R_x$ or $R_z$ rotational modes were Raman, whereas the $\nu_1$ internal, and $R_y$ rotational and $T_x$ or $T_z$ translational modes were IR inactive. Thus, $4(A_u$, $B_{1u}$, $B_{2u}$ and $B_{3u})$ $\times 3(\nu_2, \nu_3$ and $\nu_4)$ IR, and $4(A_g$, $B_{1g}$, $B_{2g}$ or $B_{3g})$ $\times 3(\nu_1, \nu_3$ or $\nu_4)$ internal Raman modes were expected. Furthermore, $4(A_u$, $B_{1u}$, $B_{2u}$ and $B_{3u})$ $\times 2(R_x$, $R_z)$ rotational or $4(A_u$, $B_{1u}$, $B_{2u}$ and $B_{3u})$ $\times 1(T_y)$ translational modes, and $4(A_g$, $B_{1g}$, $B_{2g}$ or $B_{3g})$ $\times 1$ $(R_y)$ rotational and $4(A_g$, $B_{1g}$, $B_{2g}$ or $B_{3g})$ $\times 2$ $(T_x$, $T_z)$ translational modes were expected in the IR and Raman spectra, respectively (Figure S3).

### 2.3.3. The Co$^{III}$ and I$^-$ ions in Compound **1**

Crystallographically, all Co$^{III}$ ions were identical in compound **1**, therefore, the total number of the external modes for the Co$^{III}$ ions was equal to $4 \times 3 = 12$ translational degrees of freedom. All g-modes were Raman and the $B_{1u}$, $B_{2u}$ and $B_{3u}$ modes were IR active (Figure S4) (the same logic was applicable for the iodide ions in compound **1**).

Summarizing the internal vibrations, 96 internal vibrations were obtained due to three types of ammonia molecules plus 24 internal vibrations of the four carbonate anions, altogether 120 internal vibrations. There were 48 hindered translations of the three types of ammonia molecules plus 12 hindered translations of the 4 carbonate anions plus 12 hindered translations of Co$^{3+}$ cations plus 12 hindered translations of I$^-$ anions, a total of 84 hindered translations. There were also 48 hindered rotations (librations) of the ammonia plus 12 librations of the carbonate anions, 60 librations in toto. These gave a total number of 264 vibrational degrees of freedom, equal to 22 atoms in the formula unit $\times$ 4 (number of formula units in the unit cell) $\times$ 3 (degrees of freedom per atom) (or $88 \times 3 = 264$). Out of the total number of hindered translations (84) there were three acoustic modes (of $B_{1u}$, $B_{2u}$ and $B_{3u}$ symmetry). The other 81 were true hindered translational modes.

Compound **1** was orthorhombic with space group *Pnma*, the coordinated ammonia ligands formed hydrogen bonds with the coordinated carbonate and the iodide counter-ions. The hydrogen bonds between the ammonia and carbonate ions caused symmetry lowering compared with the ideal free ammonia ($C_{3v}{\rightarrow}C_1$, i.e., $C_{3v}{\rightarrow}C_s$) and carbonate ion ($D_{3h}{\rightarrow}C_s$). The cis-$O_2CoN_4$ skeleton of the complex cation in compound **1** had distorted octahedral geometry with four different coordinated ammonia molecules and a carbonate ion. A standard approach is to classify the vibrations of the skeleton as having $C_{2v}$ symmetry, which is an oversimplification, of course. The coordinated carbonate ion occupied two neighboring coordination sites, and the residual sites were occupied with coordinated $NH_3$ ligands. There were 15 IR and Raman active ($A_1$, $B_1$, $B_2$) modes belonging to the cis-$O_2CCoN_4$ skeleton, the $A_2$ modes were only Raman active. Stretching and bending modes, three kinds for each, (NCoN, NCoO and OCoO), existed for a cis-$O_2CoN_4$ octahedron. These modes were considerably coupled [43]. The experimental and calculated band positions in the spectra of compound **1** and of the isolated $[Co(NH_3)_4CO_3]^+$ ion ($f_{CoO} = 1.25$;

$f_{CoN}$ = 1.6 [43]), respectively, are given in Table S5. Note that this approach (approximate $C_{2v}$ symmetry) was used for the skeletal modes, as well as on a few occasions later in this paper. In such a way, the results of the assignment were more directly comparable with those of previous works on such compounds [1,12,24].

These assignments were based on the general assumption that the stretching frequencies of M–N bonds were higher than those of the M–O bonds. Therefore, the experimental wavenumbers of the $MN_2$, cis-$MO_2$ and MON groups stretching vibrations were expected to follow the $\nu_s$(NMN) > $\nu_s$(NMO) > $\nu_s$(cis-OMO) order. Due to the cis-configuration of the coordinated carbonate ion, the $CoO_2$ moiety had $C_{2v}$ symmetry in compound **1**, thus, the antisymmetric stretching band ($\nu_{as}$(cis-$CoO_2$) at 392 and 397 cm$^{-1}$ in the IR and Raman spectra, respectively) was expected at higher wavenumber than that of the symmetric cis-$CoO_2$ stretching (325 and 323 cm$^{-1}$, in the IR and Raman spectra, respectively) [43–45] (Figure 4).

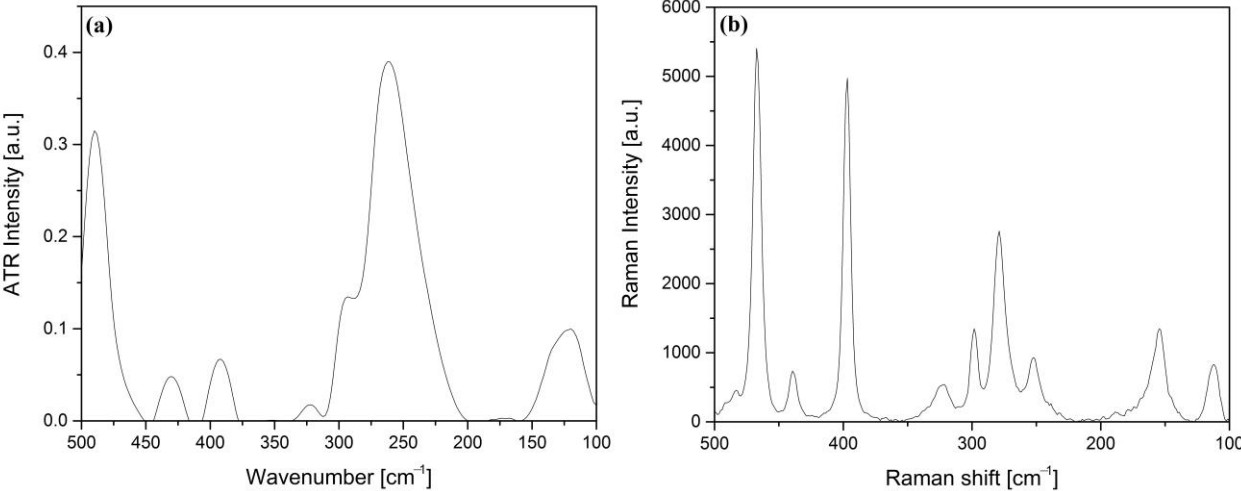

**Figure 4.** (**a**) The far-IR (25 °C) and (**b**)low-temperature Raman (785 nm excit., −150 °C) spectra between 500 and 100 cm$^{-1}$.

The Co-N modes could not be distinguished easily due to the presence of cis- and trans-$CoN_2$ and cis- and trans-CoON linkages. Furthermore, the differences in the Co-O and Co-N bond lengths, and the mixed character (Co-O+Co-N) of these bands caused a challenge in their assignment. The detailed assignments of skeletal modes are given in Table S5.

### 2.3.4. The Vibrational Modes of the Ammonia and Carbonate Ion Ligands in Compound **1**

The band assignments of ammonia ligands in the spectra of compound **1** are given in Table S6 in comparison with the data given by Siebert [38] and Goldsmith [43]. All these bands in the spectra could be assigned.

The band positions agreed very well with the results of Goldsmith [43], whereas the doubly degenerate $\delta_{as}$(HNH) mode found by Siebert [38] at the wavenumber 1603 cm$^{-1}$ was assigned to the C=O carbonyl stretching band of very close position (1595 cm$^{-1}$) [43]. It was obvious that the bands $\delta_{as}$(HNH) and $\nu_{as}$(C=O), or $\varrho$($NH_3$) and $\pi$($CO_3$) in the spectrum of compound **1** recorded by Siebert [38], were not separated well. A similar phenomenon was found in the case of the [carbonatotetraamminecobalt(III)]sulfate, where the coinciding bands could be resolved by deuteration [24]. The low intensity deformation band of $\delta_{as}$(HNH) in the IR spectrum of compound **1** was located at 1665 cm$^{-1}$, and separated well from the very intense $\nu_{as}$(C=O) band found at 1595 cm$^{-1}$. Similarly, no overlap was observed between the bands belonging to the symmetric $\delta_s$(HNH) bending mode (1316 cm$^{-1}$) and the symmetric C-O stretching modes (1283 and 1263 cm$^{-1}$) in the IR spectrum of compound **1**. These two bands overlapped in the IR spectrum of the ana-

log sulfate compound, and after deuteration the higher wavenumber component was assigned to the ammonia and the lower wavenumber components to the C-O modes. Since the $\delta_s(NH_3)$ band of compound **1** was well separated, it was easy to determine the basicity (bond strength) of ammonia in this structure with the method developed by Grinberg [1,4,12]. The results are given in Table S7.

The calculated relative bond strength showed only ±2% with changing the number of ammonia molecules or incorporation of the carbonate ion into the coordination sphere in mono- or bidentate mode (Table S4). This showed that other important factors such as the number and strength of hydrogen bonds of ammonia with carbonate oxygens or iodide ions may compensate for or overwrite the influence of the carbonate ion.

There were no remarkable differences between the Raman spectra recorded at room temperature and −150 °C. Meanwhile, the intensities of the bands belonging to the normal modes of ammonia and carbonate ligands ($A_1$, $B_1$ and $B_2$) were different in the IR and Raman spectra (Figure 5). The intensity of bands belonging to the C=O (non-bound) stretching mode in the Raman spectra was very low, whereas it was the most intense band in the IR spectrum of compound **1**. The 785 nm laser excitation did not produce NH bands intense enough to evaluate, whereas the 532 nm excitation resulted in a Raman spectrum with the bands of NH stretching and deformation modes. The bands of stretching modes (symmetric and antisymmetric) strongly overlapped (Table S5).

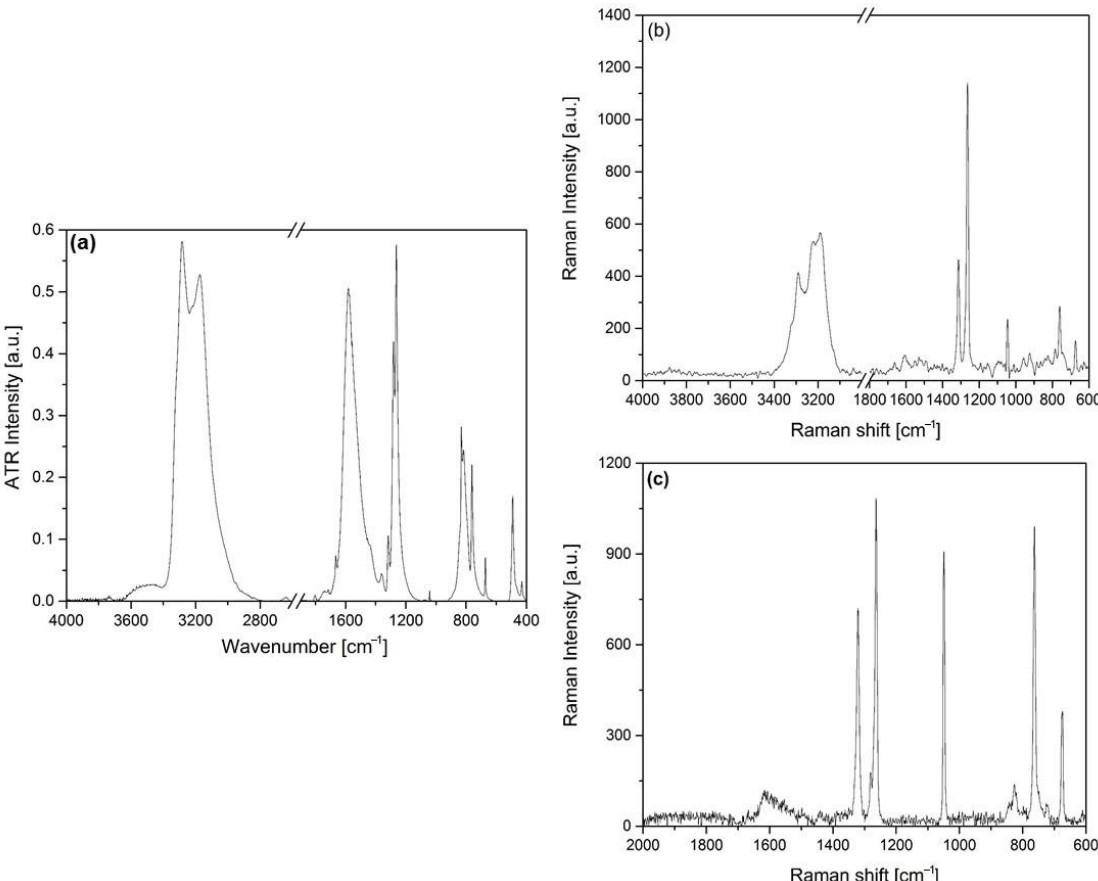

**Figure 5.** The (**a**) IR (25 °C) and low-temperature Raman spectra (−150 °C) in the region of (**b**) the ammonia (532 nm excitation); and (**c**) carbonate (785 nm excitation) vibrational modes.

The modes of the coordinated carbonate ion in compound **1** and their assignments are given in Table S8. The stretching modes of the non-coordinated carbonate oxygen ($\nu_1$) and the symmetric and antisymmetric modes of the coordinated ones ($\nu_2(A_1)$ and $\nu_4(B_1)$, respectively) could be distinguished. The highest wavenumber of the stretching modes of the carbonate ion (1617 cm$^{-1}$) belonged to the non-coordinated C=O bond. There were two

in-plane OCO deformation modes with ($\nu_5(B_1)$) and without ($\nu_3(A_1)$) involved non-bonded oxygens. The splitting of the antisymmetric stretching mode (E in $D_{3h}$) into $\nu_1(A_1) + \nu_4(B_1$ in $C_{2v}$) (Figure 5), which is characteristic for chelate-forming bidentate ligands, may have been responsible for a band at 328 cm$^{-1}$.

### 2.3.5. UV Spectroscopy

The low-spin (diamagnetic) Co$^{III}$ ion electron configuration was $t_{2g}^6$, transforming as $^1A_{1g}$. Excitation of an electron from the $t_{2g}$ orbital to the $e_g$ orbital resulted in a $t_{2g}^5 e_g^1$ configuration, which spanned $^3T_{1g} + {}^1T_{1g} + {}^1T_{2g} + {}^3T_{2g}$ states. The two triplet states lay at lower energy levels than the two singlet ones. There were four transitions found experimentally for the [Co(NH$_3$)$_4$CO$_3$] cation at 230, 300, 358, and 524 nm [46]. The most intense band in the visible region was the $^1A_{1g}{\rightarrow}^1T_{1g}$ transition and the $^1A_{1g}{\rightarrow}^1T_{2g}$ transitions were found as shoulders at 569 nm and 374 nm, respectively (Figure S5, Table S9). A shoulder was also found at 524 nm. The L($\pi$)$\rightarrow$Co($e_g$) MLCT band was assigned on the basis of the linear relationship between the singlet transitions wavenumbers and CT band positions given in [47], which suggested that the carbonate ion in compound **1** achieved > 50% covalency [48]. This demonstrated the role of the hydrogen bonds, which depends on many factors, from the geometry to the counter anion.

Two spin-allowed transitions were found in the UV-VIS spectra of compound **1** centered at 687 and 834 nm (Figure S5). There were many bands in the system, and Sastri concluded that there is significant mixing of $t_{2g}^5 e_g^1$ and $t_{2g}^4 e_g^2$ levels (these $e_g$ levels have $\sigma^*$ character) in the carbonatotetraamminecobalt(III) complexes [48].

### 2.4. Thermal Analysis of Compound **1**

The thermal decomposition of compound **1** was studied in helium and in air to investigate the influence of the external oxygen source over the decomposition processes. The main parameters of the decomposition process are summarized in Table S10. The decomposition was an endothermic single-step process in helium atmosphere while involving two partly coinciding stages—an endothermic and an exothermic—in air (Figure 6). The initial decomposition of compound **1** started at the same temperature both in helium and in air, thus, the external oxygen had no role in the first stage of the decomposition process. However, the intermediate that formed in the first decomposition stage reacted with the oxygen content of the air—resulting in heat evolution.

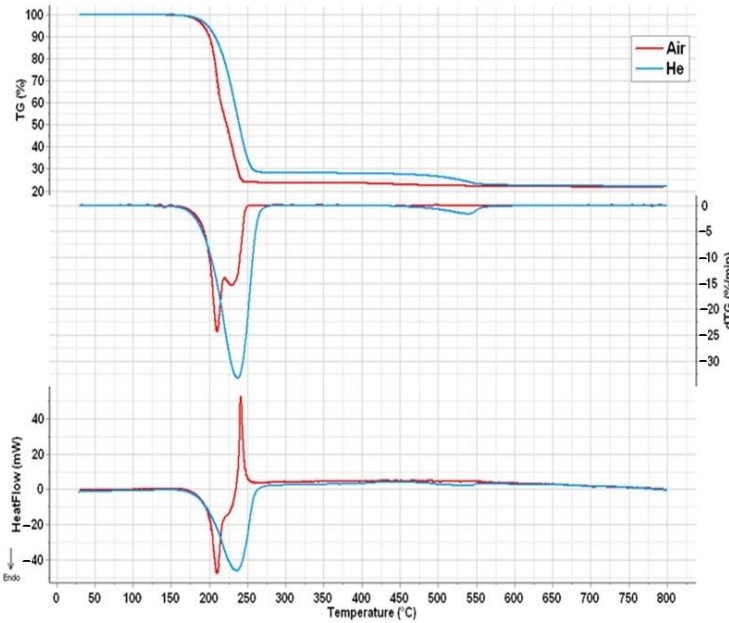

**Figure 6.** The thermal decomposition characteristics of compound **1** in helium atmosphere and in air.

The tiny metallic cobalt particles were also visible on the stereomicroscopic image taken on the solid residue, recovered after the thermal measurement in helium. They appeared as a few micrometers small, shiny, glowing particles on the surface of brown CoO particles resembling wood fibers (Figure 7).

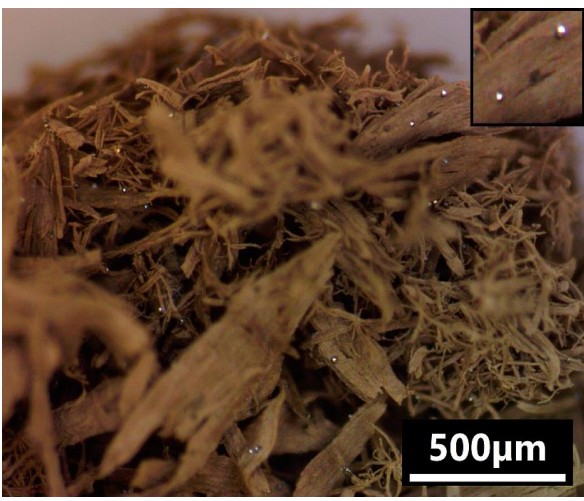

**Figure 7.** Stereomicroscopic image of the small metallic cobalt particles on the surface of CoO particles (magnification 55×).

The spot-like arrangement of cobalt particles strongly suggested that the reduction of $Co_3O_4$ (CoO) into metallic cobalt occurred with the participation of reducing components that were present as inclusions and, therefore, they produced cobalt particles locally.

In helium atmosphere, the overall mass loss of compound **1** until 800 °C was ~78%. The final decomposition product was identified as CoO and cubic metallic cobalt by XRD (Figure 8). The theoretical mass losses for the formation of CoO and metallic cobalt were ~76.1% and −81.2%, respectively. In air, the end-product of the decomposition was $Co_3O_4$, the same as the intermediates found at 300 °C in both atmospheres (XRD, Figure 8).

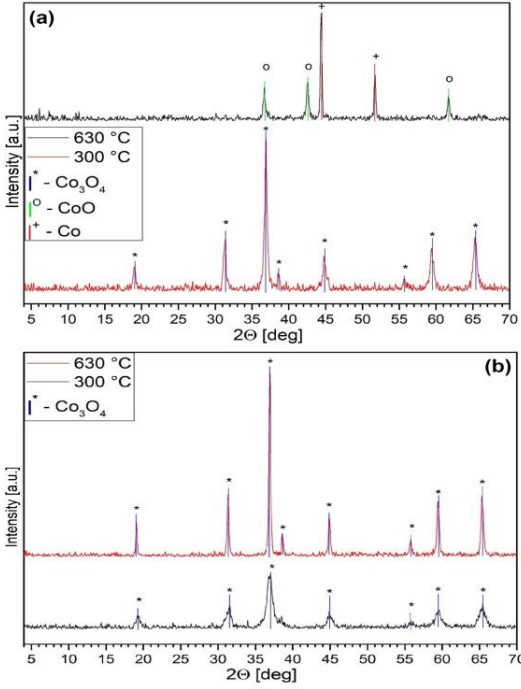

**Figure 8.** XRD of the decomposition products of compound **1** at 300 °C and 630 °C in (**a**) helium atmosphere and in (**b**) air.

The mass loss of the decomposition step in the He atmosphere ($-69.7\%$) was less than the calculated value for the $Co_3O_4$ formation ($-74.5\%$). This showed the presence of (an) X-ray amorphous material(s), which might be $CoCO_3$, or according to the IR results, compounds containing N-H and C=O groups (ammonium salts, or amides such as urea [2,49]) (Figure S6). Ammonium-ion-containing intermediates were previously found among the thermal decomposition products of [hexaamminecobalt(III)] chloride and bromide [35,50], whereas the formation of HNCO/HOCN was observed during the thermal decomposition of $[Fe(urea)_6](MnO_4)_3$ between 120 and 180 °C [2], strongly suggesting the in situ formation of urea in the reaction of $CO_2$/carbonate and $NH_3$.

The intensity of the IR bands belonging to C-N multiple bonds in the spectra of the solid decomposition intermediates that formed at 300 °C from compound **1** in air was less than that in the spectra of the intermediate formed in helium atmosphere (Figure S6). This showed that the source of C-N compounds was probably a decomposition reaction of $[Co(NH_3)_4CO_3]$, and that the oxidation decreased the amount of the organic (C-N bond containing) reaction products.

The second decomposition step of compound **1** observed in He atmosphere at 575 °C peak temperature with 8.0% mass loss and 11.61 kJ/mol reaction heat was due to the reduction of $Co_3O_4$ to CoO and metallic cobalt (Figures 6 and 8). The thermal decomposition of $Co_3O_4$ into CoO proceeded at a temperature higher than 575 °C [51], thus, the formation of CoO and metallic Co could be attributed to a reduction process with the involvement of the N-H bond containing amorphous residues. This reduction reaction could not be completed to form only metallic cobalt due to the insufficient amount of reducing components. The morphological consequences of this reduction can be visualized by comparing the SEM pictures of the decomposition intermediate (300 °C, $Co_3O_4$, Figure 9a,b) and that of the final decomposition product (630 °C, Co + CoO, or $Co_3O_4$, Figure 9c,d).

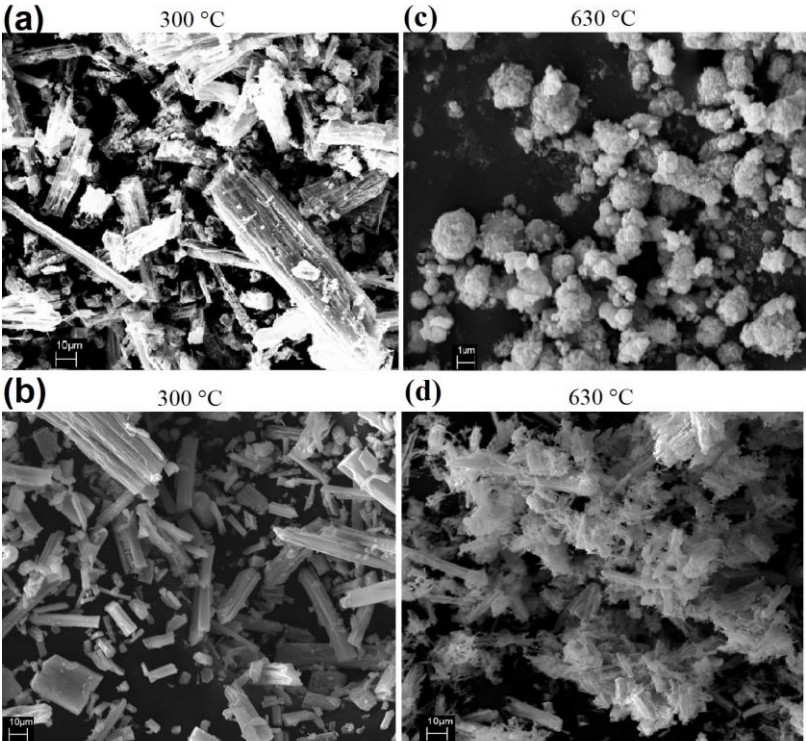

**Figure 9.** The SEM pictures of the decomposition products formed from compound **1** at 300 and 630 °C: (**a**,**c**) in air, and (**b**,**d**) in He atmosphere).

The TG-MS studies on the decomposition of compound **1** in He atmosphere showed that the main redox reactions proceeded in one complex decomposition step. The peak temperature of the total ion-chromatogram was 239 °C. A simultaneous formation of

$CO_2$, $H_2O$, $NH_3$, $N_2$, and HI was found in the complex reaction steps. Small amounts of $NO_2^+$ ($m/z = 46$) and $NO^+$ ($m/z = 30$) were also detected. Two weak peaks at $m/z = 127$ and 128 were also observed. Iodine is a monoisotopic element ($M = 127$), thus, the $m/z = 127$ belonged to the iodine atom, whereas the $m/z = 128$ could only be $HI^+$ (Figures 10 and S7).

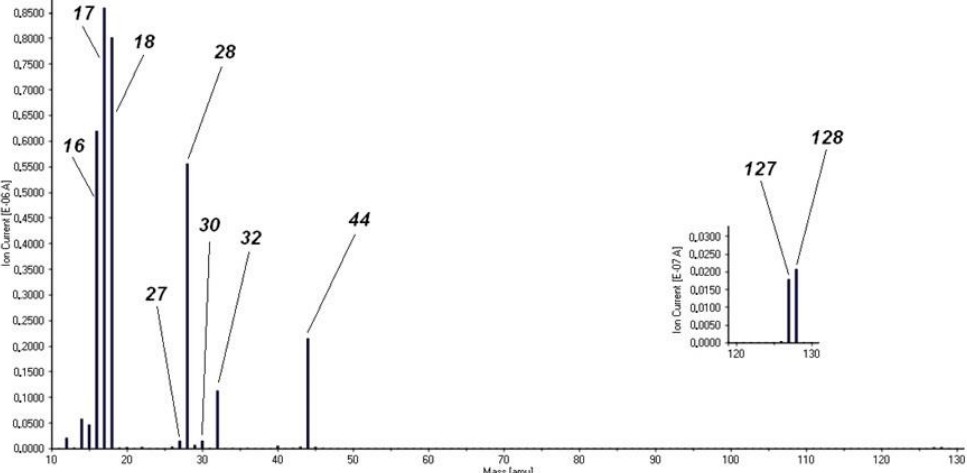

**Figure 10.** Analogous TG-MS spectra of the main gaseous decomposition products of compound **1** (please note that the numbers are representing $m/z$ data).

The appearance of a water ($m/z = 18$) peak confirmed the redox reaction between the ammonia and carbonate ions ($NH_3$ as the hydrogen source and carbonate ion*s* as the oxygen source), which was confirmed by the appearance of peaks belonging to carbon–nitrogen containing components as $HCN^+/HNC^+$ ($m/z = 27$), $CN^+$ ($m/z = 26$) $HOCN^+/HNCO^+$ ($m/z = 43$) or $OCN/CNO^+$ ($m/z = 42$) (Figure 10).

The $m/z = 28$ peak may have been due to $N_2^+$ and $CO^+$ (the latter a $CO_2^+$ fragmentation product), therefore the intensity ratios of 44/30 ($CO_2^+ + N_2O^+/NO^+$) and 44/28 ($N_2O^+ + CO_2^+/CO^+ + N_2^+$) were compared, which showed that the main component of the $m/z = 28$ peaks was the $N_2^+$. Comparison of the intensity ratios of the peaks with 18/17 ($H_2O^+/OH^+ + NH_3^+$), 17/16 ($NH_3^+ + OH^+/NH_2^+ + O^+$), 16/15 ($NH_2^+ + O^+/NH^+$) and 15/14 ($NH^+ + NO^{2+}/N^+$) [52] showed that the main component of $m/z = 17$ and 44, were ammonia and carbon dioxide, respectively.

The $m/z = 254$ ($I_2$) was outside of our detection range, thus, formation of molecular $I_2$ could not be followed by TG-MS, however, Wendland observed iodine vapor formation [33], thus, monoatomic iodine may have been the fragment ion of $HI^+$ and $I_2^+$ as well. The HI may also have been obtained from the reaction of elementary iodine with the ammonia ligands. We heated compound **1** under isotherm conditions at around ~200 °C, and the IR spectra (Figure S8) and XRDs (Figure S9) of the intermediate (and its subsequent decomposition products) before and after aqueous washing were recorded. The mass loss at isotherm heating of compound **1** at 200 °C in air resulted in 67.4% mass decrease. Wendland found that only 77% and 60% of the expected ammonia and carbon dioxide content of compound **1** were released in gaseous state, respectively [32], and the elimination of 0.5 mol of $I_2$, $4 \times 17 \times 0.77$ mol of $NH_3$, and $44 \times 0.60$ mol of $CO_2$ from one mol of compound **1** resulted in 65.9% mass loss, which agreed well with each other. In order to see an interim situation during the thermal decomposition, the residue was leached with water. Due to the aqueous leaching, the mass loss was 2.78%. There was no compound **1** observed after the evaporation of the aqueous leachate, thus, it could be concluded that the vast majority of compound **1** decomposed during the 1 h thermal treatment at 200 °C (Figure S9). These results confirmed that the main part of the carbonate content was eliminated in the form of CO, or a C-N compound such as urea or biuret. Urea and biuret decompose in situ above 132.7 and 190 °C, respectively.

The overall heats of reaction in helium and in air were found to be 26.7 and 162.1 kJ/mol, respectively. In air, the endothermic stage showed 219.0 kJ/mol heat of reaction that was followed by an exothermic reaction with −56.9 kJ/mol heat of reaction. This showed that despite the identical initial steps in both atmospheres, the subsequent decomposition reactions of the $[Co(NH_3)_4CO_3]$ intermediate were very different in the presence or absence of oxygen. In the presence of oxygen, before completing the first endothermic ammonia loss process, the shape of the DSC curve changed, and an exothermic process occurred (Figure 6). The heat released was attributed to an oxidation process. The main reaction product in both atmospheres at 300 °C was $Co_3O_4$ (Figure 8). In an inert atmosphere, $Co_3O_4$ is reduced into CoO + Co around 575 °C, while in the presence of air, the reducing components start to combust even at around ~230 °C. The IR and Raman spectra of the decomposition intermediates that formed at 300 °C in helium and air, and the final product that formed at 630 °C in air, confirmed the presence of $Co_3O_4$ (Figures S10 and S11). The intense IR bands at 570 and 661 cm$^{-1}$ belonged to the $\nu_{Co\text{-}O}$ modes of Co$^{III}$ ions located at octahedral sites and Co$^{II}$ ions located at tetrahedral sites of the spinel lattice, respectively [51–54]. The Raman spectra showed the expected Raman bands of $Co_3O_4$ at 483 cm$^{-1}$ ($E_g$), 524 cm$^{-1}$ and 622 cm$^{-1}$ ($F_{2g}$), and 694 cm$^{-1}$ ($A_{1g}$). In air, no CoO formed at 630 °C (no Raman bands of CoO were observed at 675 and 455 cm$^{-1}$) in accordance with the thermal oxygen desorption curve of the CoO + $O_2$ system, which shows the oxidation of CoO into $Co_3O_4$ with oxygen uptake above 200 °C [51]. There were only minor morphological differences for $Co_3O_4$ formed in air upon heating at 300 and 630 °C as shown in Figure 8.

Our TG-MS, XRD, IR, and Raman, as well as Wendlandt's magnetic susceptibility measurement results about the thermal decomposition of compound **1,** allowed us to conclude as follows:

(1) The initial thermal decomposition step of compound **1** proceeded the same way both in inert atmosphere and in air (Figure 6) and consisted of the solid-phase quasi-intramolecular oxidation of iodide ion with Co$^{III}$:

$$[Co(NH_3)_4CO_3]I(s) = [Co(NH_3)_4CO_3](s) + 1/2I_2\ (s) \tag{1}$$

The character of the mass loss observed [35] may be attributed to the sublimation of elementary iodine, and it became more intensive only around the boiling point (184.3 °C) of iodine. This could explain why Wendland found a lack of regular relationship between the mass loss and the Co$^{II}$ content during the thermal decomposition of compound **1**, since Co$^{III}$ did not oxidize bromide and chloride ions in compounds **2** and **3** into elementary halogens, but the oxidation of ammonia by Co$^{III}$ was accompanied by $Co_2OX_2$ (X = Br, Cl) formation, and the Co$^{II}$ content was found to be proportional to the mass loss in these reactions [33];

(2) The subsequent decomposition of the intermediate $[Co(NH_3)_4CO_3]$ was not the same in inert and in oxygen-containing atmosphere. Cobalt(II) carbonate does not form $[Co(NH_3)_4CO_3]$ or other complex in an $NH_3$ stream [55], and the only known ammine complex of cobalt(II) carbonate, $Co(NH_3)_3CO_3\cdot4H_2O$ prepared in aq. ammonia solution is very unstable even at room temperature. Thus, $[Co(NH_3)_4CO_3]$ was expected to be a metastable compound at 200 °C. In the absence of air, ligand loss reactions and interactions between the $NH_3$ and $CO_2$ (carbonate) components resulted in the formation of C-N bonds containing intermediates. Since the oxidation number of cobalt in $[Co^{II}(NH_3)_4CO_3]$ increased during the $Co_3O_4$ formation ($Co^{II}Co^{III}_2O_4$) in the oxygen-free environment, 25% of the oxygen content of $Co_3O_4$ had to originate formally from carbon dioxide $[Co^{II}(NH_3)_4CO_3 = Co^{II}O + 4NH_3 + CO_2]$. This oxygen transfer should be accompanied by the formation of carbon compounds with reduced oxygen content (the O/C ratio should be <2, because the O/C ratio is 2 in $CO_2$). Based on the TG-MS, XRD and IR results (Figures S7–S9) the intermediate $CoCO_3$ may be decomposed into $Co_3O_4$ and CO [56], or a quasi-intramolecular redox reaction of $[Co(NH_3)_4CO_3]$ may result in the formation of the C-N bond containing intermediates as urea, biuret or other amides, which in situ transforms into HNCO/HOCN type

decomposition products at ~200 °C (the decomposition points of urea and biuret are 132.7 °C and 190 °C, respectively). The amount of organic compounds formed in inert atmosphere is enough to reduce $Co_3O_4$ into CoO and partly into metallic Co (Scheme 1).

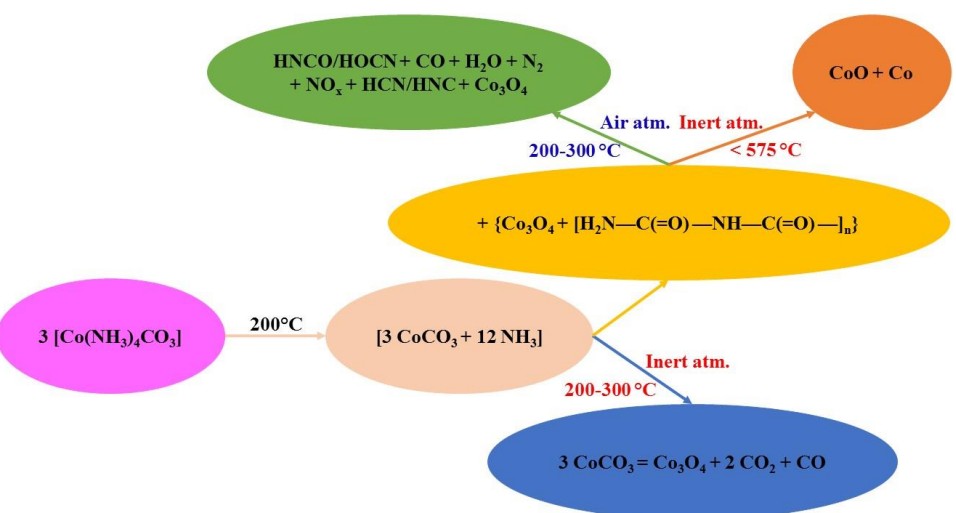

**Scheme 1.** Reaction routes during the thermal decomposition of $[Co(NH_3)_4CO_3]$.

(3) In the sample prepared at 200 °C in air, we could not detect crystalline $CoCO_3$ (Figure S9); only badly crystallized $Co_3O_4$ was found. IR studies, however, showed the presence of other X-ray amorphous components (ESI Figure S8). The strongly overlapped bands may belong to C(=O)-NH, ammonium ion and carbonate ion containing materials, and the aqueous leaching left a water-insoluble residue showing carbonate ion and coordinated hydroxide ion bands ($\nu_{OH}$ = ~1055 cm$^{-1}$). The IR spectrum of this residue was similar to the IR spectrum of basic cobalt carbonate [56], and based on the intensity ratios of the carbonate (~1400 cm$^{-1}$) and hydroxide ion signals (~1050 cm$^{-1}$), a hydroxide-ion rich basic cobalt carbonate formed from the decomposition intermediate prepared at 200 °C. The aqueous leaching resulted in removing the water-soluble compounds that showed coinciding bands with the carbonate ion.

For an insight into the mechanism of the solid-phase oxidation processes in compounds **1–3**, a systematic complex study of the thermal decomposition of compounds **1–3** and their fluoride analog [22], $[Co(NH_3)_4CO_3]F$, is planned, where the fluoride ions cannot be oxidized at all by the $Co^{III}$ or air, thus, the $Co^{III}$- and air-ammonia interactions can be separated from each other and from halogenide oxidation processes.

### 3. Materials and Methods

The chemical grade starting materials and the analytical reagents were supplied by Deuton-X Ltd., Érd, Hungary. The determination of the formula of compound **1** was performed with gravimetric analysis described in detail in our earlier papers [1,24]. The iodide content was measured as AgI after the deposition of silver iodide with the addition of silver nitrate solution. The ammonia content was measured after the separation of iodide content as AgI to avoid iodine formation and reaction of the liberated iodine with ammonia, by boiling with 10% aq. NaOH solution and absorbing the $NH_3$ gas in $H_2PtCl_6$ solution then measuring the mass of the $(NH_4)_2PtCl_6$ precipitate. The sodium carbonate solution formed during the alkaline boiling was reacted with 1 M HCl, and the evolved carbon dioxide was reacted with a saturated barium hydroxide solution and measured as barium carbonate. The cobalt content was determined as $Co_3O_4$ after treatment of the mother liquor with sodium peroxydisulfate and NaOH, and the formed brown precipitate was measured after heating at 950 °C for 2 h [24]. Isotherm heating of compound **1** was performed under He and air atmospheres between 200 and 950 °C for 1 or 2 h.



### 3.1. Vibrational Spectroscopy

The far−IR and mid−IR spectra of compound **1** were recorded in an attenuated total reflection mode (ATR). The details of the instrumentation (BioRad−Digilab FTS−30−FIR and Bruker Alpha IR spectrometers) were given in [18]. The far-IR and FT-IR measurements were carried out between 400–40 cm$^{-1}$ and 4000–400 cm$^{-1}$, respectively.

Raman spectroscopic measurements were performed at 298 and 123 K between 2000 and 100 cm$^{-1}$ and 4000 and 100 cm$^{-1}$, with 785 nm diode laser and 532 nm Nd: YAG laser external sources, at ~80 and ~40 mW powers, respectively. The details of the instrumentation (Horiba Jobin−Yvon LabRAM microspectrometer, Olympus BX−40 optical microscope, a Linkam THMS600 stage) were given in [57–59]. The laser beams were focused on an objective of 20×. Compound **1** was sensitive enough to local heating caused by the laser, therefore, D1 intensity filters were inserted into both laser beams to decrease the laser power to 10%. A confocal hole of 1000 µm, and monochromators with 950 and 1800 groove mm$^{-1}$ gratings for the diode and the Nd:YAG laser were used for light dispersion, respectively. The resolution was 4 cm$^{-1}$, and the exposure times were 60 s.

### 3.2. UV−Vis Spectroscopy

The UV−VIS spectra were recorded in diffuse reflectance mode, the details of the instrumentation were given in [9] (Jasco V−670 UV–VIS instrument, an NV−470 integrating sphere, $BaSO_4$ as standard).

### 3.3. Scanning Electron Microscopy

SEM measurements were conducted according to the method given in [13] with the use of a Zeiss EVO40 microscope (20 kV acceleration voltage).

### 3.4. Powder X-ray Diffractometry

Powder X-ray tests were performed with a Philips (Amsterdam, The Netherlands) PW−1050 Bragg–Brentano parafocusing goniometer equipped with a copper cathode (40 kV, 35 mA, secondary beam graphite monochromator, proportional counter). Scans were recorded in step mode, and the diffraction patterns were evaluated with a full profile fitting technique [57,58].

### 3.5. Single Crystal X-ray

The single crystal X-ray diffraction measurement on compound **1** was performed at 303.46 K. Intensity data were collected on an XtaLAB Synergy R, HyPix diffractometer (rotating anode X-ray tube, PhotonJet R X-ray source, mirror monochromator; Cu-$K_\alpha$ radiation, λ = 1.54184Å) [2]. The cell parameters were determined by least-squares method, and analytical absorption correction was applied to the data. The structure was solved with the intrinsic phasing method as implemented in SHELXT [60]. Refinement was carried out using SHELXL [61] and Olex2 [62] with full matrix least-squares method on $F^2$. Hydrogen atoms were included in the structure factor calculation, but they were not refined. The isotropic displacement parameters of the hydrogen atoms were approximated from the $U$(eq) value of the atom they were bonded to. Olex2 [62] and Mercury [63] were used for molecular graphics and analyzing crystal packing. Crystal data and details of the structure refinement are listed in Tables S1–S5. The CCSD number was 2235101.

### 3.6. Thermogravimetric Analysis

Simultaneous thermogravimetric analysis (TGA), differential scanning calorimetry (DSC) and mass spectrometric evolved gas analysis (MS-EGA) were performed with the use of a SETARAM Labsys Evo and a Pfeiffer Vacuum OmniStar instrument in a He (6.0) gas stream with a flow rate of 90 mL min$^{-1}$. The measurements were recorded in the 25–800 °C temperature range, at a heating rate of 20 °C min$^{-1}$ in a 100 µL alumina crucible. The obtained data were evaluated by Calisto Processing software, ver. 2.092. The analysis of the evolved gases/decomposition products was carried out on a Pfeiffer

(Aßlar, Germany) Vacuum OmniStar™ quadrupole mass spectrometer coupled to the above-described TGA-DSC instrument. The gas splitters and transfer lines to the spectrometer were thermostated to 210 °C. The measurements were taken in SEM bar graph cycles acquisition mode. The total ion currents (TIC), the analog bar graph spectra and the separate ion current of each scanned individual mass (121 masses) were recorded. The scanned mass interval was 10–130 amu, with a scan speed of 20 ms amu$^{-1}$, and the spectrometer was operated in electron impact mode. Low-temperature DSC study was performed in inert atmosphere between −150 and 25 °C in a Perkin Elmer DSC 7 (Perkin-Elmer, Waltham, MA, USA) instrument.

*3.7. [Carbonatotetraamminecobalt(III)] Nitrate Hemihydrate*

An amount of 10 g of cobalt(II) nitrate hexahydrate was dissolved in 10 mL of hot water and added to 20.0 g of ammonium carbonate dissolved in a mixture of 50 mL of 25% aq. ammonia and 100 mL of water. An amount of 25 mL of 3.0% aq. $H_2O_2$ was slowly added to the mixture. After 10 min stirring, the volume of the solution was reduced with evaporation to 25 mL while adding 0.5 g of ammonium carbonate during the procedure 10 times (altogether 5 g) and then filtered. The solution was cooled in an ice bath, the crystals were filtered off, and washed with abs. ethanol. The yield was 57.4%. The hemihydrate of the [carbonatotetraamminecobalt(III)] nitrate was confirmed by powder XRD [64].

*3.8. [Carbonatotetraamminecobalt(III)] Iodide (Compound **1**)*
3.8.1. Procedure 1

Compound **1** was precipitated from a solution of 2 g of [carbonatotetraamminecobalt(III)] nitrate hemihydrate dissolved in 50 mL of cold water with the addition of 5 g of solid ammonium iodide as a carmine-red precipitate of small needles. The crystals were washed with a mixture of 1 volume of water plus 3 volumes of 96% ethanol, yielding 89.0%.

3.8.2. Procedure 2

An amount of 20 g of cobalt carbonate was dissolved in the required amount of dilute (10%) hydroiodic acid, and the filtrate was concentrated down to a 100 mL volume with boiling and poured into a mixture of 250 mL of 25% aq. ammonia and 200 g of ammonium carbonate dissolved in 1 L of water. This mixture was oxidized by bubbling air for 4 h. The solution was evaporated to 500 mL with boiling with intermittent adding (4–5 times, 5 g each) of ammonium carbonate. After cooling down, the brownish-violet needle-like crystals were separated. The crystallization of mother liquor resulted in another crop of compound **1**, and the overall yield was 76%.

**4. Conclusions**

1. [$\kappa^2$-O,O′-Carbonatotetraamminecobalt(III)] iodide, ([Co(NH$_3$)$_4$CO$_3$]I), compound **1** was prepared with 89% yield. The correlation analysis showed three kinds of spectroscopically different ammonia ligands in compound **1**. All normal modes in the vibrational spectra (IR and Raman) and the UV bands of compound **1** were assigned;

2. Compound **1** was orthorhombic, and isomorphous with the analogous bromide. The distorted octahedral complex cation contained four ammonia ligands and a bidentate-coordinated carbonate anion. The carbonate ion formed a four-membered symmetric planar chelate ring. There was no observed trans-effect for the carbonate ion on the equatorial Co-N distances. The complex cations and iodide ions were arranged into ion pairs and each cation bound its iodide pair through three hydrogen bonds. The complex cations were bound to each other by N-H⋯O hydrogen bonds and formed zigzag sheets via an extended 2D hydrogen bond network;

3. The thermal decomposition of compound **1** started with the solid-phase quasi-intramolecular oxidation of the iodide ion by Co$^{III}$ with the formation of [Co$^{II}$(NH$_3$)$_4$CO$_3$] and I$_2$. The intermediate Co$^{II}$-complex in situ decomposed into Co$_3$O$_4$ and C-N containing intermediates. The ammonia ligand loss resulted in CoCO$_3$,

which in situ decomposed into $Co_3O_4$ and carbon monoxide. A quasi-intramolecular solid-phase redox reaction of $[Co(NH_3)_4CO_3]$ might result in C-N bond-containing compounds with a sub-stoichiometric release of ammonia and $CO_2$ from compound **1**. In an inert atmosphere, the C-N containing compounds reduced $Co_3O_4$ into CoO and Co. In oxygen-containing atmosphere, the end-product was $Co_3O_4$, and the endothermic ligand loss reaction coincided with the consecutive exothermic oxidation processes. The carbonate ion remaining in the decomposition residues formed at 200 °C transformed into a coordinated hydroxide-ion rich basic cobalt(II) carbonate during aqueous leaching.

The formation of $[Co(NH_3)_4CO_3]$ and its decomposition into -C(=O)-NH- compounds is a promising result to initiate novel studies on the $Co^{II}$ catalyzed interaction of $NH_3$ and $CO_2$ at atmospheric pressure, targeting urea synthesis.

**Supplementary Materials:** The following supporting information can be downloaded at: https://www.mdpi.com/article/10.3390/inorganics11020068/s1, Figure S1: DSC results of compound **1** between −150 °C and room temperature; Figure S2: (a) The calculated (from SXRD data) and (b) the experimental powder X-ray diffractogram of compound **1**; Figure S3: Factor group analysis of (a) internal and (b) external $CO_3^{2-}$ modes in $[Co(NH_3)_4CO_3]I$; Figure S4: Factor group analysis of hindered translations of monoatomic structural motifs ($Co^{3+}$ or $I^-$) in $[Co(NH_3)_4CO_3]I$; Figure S5: The UV-VIS spectra of compound **1** in (a) 200–440 nm range, (b) 420–650 nm range, and (c) 660–900 nm range; Figure S6: The IR spectrum of the decomposition intermediates formed at 300 °C in air, recorded in the range of 700–3700 $cm^{-1}$ (enlarged part of the IR spectra of the decomposition intermediates formed in He (a) and in air (b), in the N-H and -C(=O)-NH- groups absorption bands range); Figure S7: The TG-MS ion intensity curves for curves of $m/z$ = 14, 15, 28, 30 and 44 (a), $m/z$ = 15, 16, 17, and 18 (b), $m/z$ = 26, 27, 42 and 43 (c), and $m/z$ = 127 and 128 (d) ions; Figure S8: The IR spectra of the decomposition product of compound **1** formed under isotherm conditions at around ~200 °C; Figure S9: The powder-XRD of the decomposition product of compound **1** formed under isotherm conditions at around ~200 °C: (a) compound **1**, (b) decomposition product of compound **1** under isotherm conditions at around ~200 °C, and (c) the (b) after aqueous washing; Figure S10: The Raman spectra of decomposition products of compound **1** formed at 300 and 630 °C in air; Figure S11: The IR spectra of decomposition products of compound **1** formed at 300 and 630 °C in inert atmosphere. Table S1: Crystal data and structure refinement details of $[Co(NH_3)_4CO_3]I$. Table S2: Bonds lengths in the crystal structure of $[Co(NH_3)_4CO_3]I$ (Symmetry codes to generate equivalent atoms: 1. [8_565] x,-y-1/2+1,z). Table S3: Bonds angles in the crystal structure of $[Co(NH_3)_4CO_3]I$ (Symmetry codes to generate equivalent atoms: 1. [8_565] x,-y-1/2+1,z). Table S4: Analysis of Potential Hydrogen Bonds and Schemes with d(D···A) < R(D)+R(A) + 0.50, d(H···A) < R(H) + R(A) − 0.12 Ang., D-H···A > 100.0 Deg. Table S5: The internal vibrational modes of the cis-$O_2CoN_4$ skeleton and their assignments in the far-IR and Raman spectra of compound **1** (assuming effective $C_{2v}$ symmetry of the skeleton). Table S6: The assignment of the ammonia vibrational modes in the IR and Raman spectra of compound **1** (classified under $C_{3v}$ symmetry). Table S7: The relative Co-N donor bond strength in the amminecobalt iodide complexes. Table S8: The κ2-O,O′-coordinated (chelate-forming) carbonate ion ($C_{2v}$) vibrational modes and their assignments in compound **1** (a tentative $C_{2v}$ symmetry of the coordinated carbonate anions was assumed). Table S9: Experimental UV–Vis data for compound **1**, $[Co(NH_3)_4CO_3]_2SO_4 \cdot 3H_2O$, and the calculated data for the $[Co(NH_3)_4CO_3]^+$ ion. Table S10: The main parameters of the decomposition process of compound **1**.

**Author Contributions:** Conceptualization, L.K.; formal analysis, V.M.P. and L.B.; investigation, K.A.B., F.S., Z.D., L.T., A.F. and N.B.; resources, Z.H., K.A.B. and L.K.; writing—original draft preparation, L.K.; writing—review and editing, Z.H., K.A.B. and V.M.P.; visualization, K.A.B. and L.B.; supervision, Z.H. and L.K.; funding acquisition, K.A.B. and L.K. All authors have read and agreed to the published version of the manuscript.

**Funding:** The research was supported by the European Union and Hungary, co-financed by the European Regional Development Fund (VEKOP-2.3.2-16-2017-00013) (L.K.) and the ÚNKP-21-3 and 22-3 New National Excellence Program of the Ministry for Innovation and Technology sourced from the National Research, Development, and Innovation Fund (K.A.B.). The crystallographic study was supported by the European Union and Hungary, and co-financed by the European Regional Development Fund (VEKOP-2.3.3-15-2017-00018).

**Data Availability Statement:** Not applicable.

**Conflicts of Interest:** The authors declare no conflict of interest. The funders had no role in the design of the study; in the collection, analyses, or interpretation of data; in the writing of the manuscript; or in the decision to publish the results.

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
