# Peer review of "Structural, Spectroscopic, and Thermal Decomposition Features of [Carbonatotetraamminecobalt(III)] Iodide—Insight into the Simultaneous Solid-Phase Quasi-Intramolecular Redox Reactions"

_inorganics, doi:10.3390/inorganics11020068_

Round 1
Reviewer 1 Report
The paper present the synthesis and characterization of ([Co(NH3)4CO3]I. The compound was thoroughly characterized using various spectral techniques. Also, the authors explore the simultaneous solid-phase Quasi-intramolecular Redox Reactions. Overall the manuscript has covered a lot of aspects of basic Inorganic Chemistry, but my only concern is the author needed to explore applications of the reported compounds.
The author writes the research article in the style of a review article. I am afraid this paper needs to be shortened more.
Please improve the abstract and conclusion part as I feel it is so lengthy.
The author can also improve the single crystal x-ray description by comparing it with other reports in the literature. See some references
J Biomol Struct Dyn 2022, 40, 9067-9080, J Biomol Struct Dyn 2021, 39, 4037-4050, RSC advances 2021, 11, 16881-16891
The Figure quality is too poor in the whole manuscript.
Author Response
First of all, we would like to thank to the Reviewer to improve the quality of our manuscript. Here we answer the Reviewer’s comments and suggestions.
The paper present the synthesis and characterization of ([Co(NH3)4CO3]I. The compound was thoroughly characterized using various spectral techniques. Also, the authors explore the simultaneous solid-phase Quasi-intramolecular Redox Reactions. Overall the manuscript has covered a lot of aspects of basic Inorganic Chemistry, but my only concern is the author needed to explore applications of the reported compounds.
The part where we gave the application as starting material in the preparation of cobalt oxide catalyst has been changed (new pieces of information have been inserted) to show the role of the title compound in the preparation of a wide scale of Co-oxide and mixed oxide precursors.
The author writes the research article in the style of a review article. I am afraid this paper needs to be shortened more. Please improve the abstract and conclusion part as I feel it is so lengthy.
The abstract and conclusion were shortened. In the text, we could do some reduction regarding to the synthesis methods. Basically, we inserted those pieces of information to show the challenges in the synthesis and analysis, due to the possibility of various by-product formation (inner-sphere and aquo-or iodo-complexes, outer sphere carbonate or hydrogen carbonate complexes, penta- or hexaammines) and reaction of each component during the analysis (CoIII + iodide resulting in iodine formation, iodine + ammonia with N2 and NH4I formation). But, according to the reviewer’s request, this part is also shortened.
The author can also improve the single crystal x-ray description by comparing it with other reports in the literature. See some references
J Biomol Struct Dyn 2022, 40, 9067-9080, J Biomol Struct Dyn 2021, 39, 4037-4050, RSC advances 2021, 11, 16881-16891
There were comparisons of the structural elements of the title compound with that of the analogous bromide complex, but no other crystal structure is known with monovalent anions. Only the nitrate and perchlorate crystal structures are known, which have crystalline water. This causes the appearance of a number of new kinds of hydrogen bonds involving water, causing changes generated by the hydrogen-bond networks made by water, thus the changes were thought to be non-comparable with the anhydrous compound structures. But, according to the reviewer’s request, we inserted a discussion about these phenomena and compare the cation geometry changes in these compounds in relation to the title compound.
The Figure quality is too poor in the whole manuscript.
The quality of figures has been improved.

Reviewer 2 Report
This is a quite surprising manuscript. The works on mononuclear all-inorganic metal complexes of 3d metals were mostly performed decades ago (this is also well reflected in references). However, the authors decided to return to some of these objects, selecting iodide salt of heteroleptic ammino/carbonato complex of Co(III). Indeed, its structure was unknown and there were some unclearity with thermal decomposition pattern (Ref. 32, Wendlandt et al.)
So, the work indeed has scientific novelty, although it refers to quite old and well-studied area. Technically, all experiments are performed well and interpretation of the results is unambiguous, so I can recommend acceptance of this manuscript.
My only recommendation is related to references: maybe authors would consider these recent papers on Co complexes (this is not obligatory, suggestion only):
a) 10.1016/j.mencom.2021.09.011
b) 10.1134/S1070328422700129
c) 10.1134/S1070328422700026
d) 10.1134/S0036023622080150
e) 10.1134/S1070328422090068
f) 10.1134/S0022476622100043
g) 10.1134/S0022476622090128
h) 10.1134/S1070328420010017
Author Response
First of all, we would like to thank to the Reviewer to improve the quality of our manuscript. Here we answer the Reviewer’s comments and suggestions.
This is a quite surprising manuscript. The works on mononuclear all-inorganic metal complexes of 3d metals were mostly performed decades ago (this is also well reflected in references). However, the authors decided to return to some of these objects, selecting iodide salt of heteroleptic ammino/carbonato complex of Co(III). Indeed, its structure was unknown and there were some unclearity with thermal decomposition pattern (Ref. 32, Wendlandt et al.). So, the work v.
Yes, it is true, similar complexes were studied earlier, but not the iodide, and among the tetramminecarbonato complexes, only four (bromide, sulfate, perchlorate, nitrate) have known crystal structures. To clarify the mechanism of the decomposition reaction resulting in cobalt oxide catalysts, these studies of the hardly characterized iodide compound became important, especially, because it is a precursor to prepare new complexes and a candidate to develop new mixed oxide catalysts.
Technically, all experiments are performed well and interpretation of the results is unambiguous, so I can recommend acceptance of this manuscript.
My only recommendation is related to references: maybe authors would consider these recent papers on Co complexes (this is not obligatory, suggestion only):
- a) 10.1016/j.mencom.2021.09.011
- b) 10.1134/S1070328422700129
- c) 10.1134/S1070328422700026
- d) 10.1134/S0036023622080150
- e) 10.1134/S1070328422090068
- f) 10.1134/S0022476622100043
- g) 10.1134/S0022476622090128
- h) 10.1134/S1070328420010017
We have inserted new references.

Round 2
Reviewer 1 Report
The authors have significantly improved the quality of the research article and it can be accepted in this form.